# Structure of a AAA+ unfoldase in the process of unfolding substrate

Zev A Ripstein[1,2], Rui Huang[2,3,4], Rafal Augustyniak[2,3,4], Lewis E Kay[1,2,3,4]*, John L Rubinstein[1,2,5]*

[1]The Hospital for Sick Children Research Institute, Toronto, Canada; [2]Department of Biochemistry, University of Toronto, Toronto, Canada; [3]Department of Molecular Genetics, University of Toronto, Toronto, Canada; [4]Department of Chemistry, University of Toronto, Toronto, Canada; [5]Department of Medical Biophysics, University of Toronto, Toronto, Canada

**Abstract** AAA+ unfoldases are thought to unfold substrate through the central pore of their hexameric structures, but how this process occurs is not known. VAT, the *Thermoplasma acidophilum* homologue of eukaryotic CDC48/p97, works in conjunction with the proteasome to degrade misfolded or damaged proteins. We show that in the presence of ATP, VAT with its regulatory N-terminal domains removed unfolds other VAT complexes as substrate. We captured images of this transient process by electron cryomicroscopy (cryo-EM) to reveal the structure of the substrate-bound intermediate. Substrate binding breaks the six-fold symmetry of the complex, allowing five of the six VAT subunits to constrict into a tight helix that grips an ~80 Å stretch of unfolded protein. The structure suggests a processive hand-over-hand unfolding mechanism, where each VAT subunit releases the substrate in turn before re-engaging further along the target protein, thereby unfolding it.

*For correspondence: kay@pound.med.utoronto.ca (LEK); john.rubinstein@utoronto.ca (JLR)

## Introduction

The 83 kDa VAT protein, which is composed of an N-terminal regulatory domain (NTD) and two tandem nucleotide-binding AAA+ domains (NBD1 and NBD2), assembles into a 500 kDa homohexamer (*Figure 1—figure supplement 1*). The physiological function of the VAT NTDs remains unclear. Truncation of the first 182 residues of VAT removes the regulatory NTD, producing a construct (ΔN-VAT) that has both higher ATPase and unfoldase activity than the intact protein (*Gerega et al., 2005*; *Rothballer et al., 2007*; *Barthelme and Sauer, 2012*). In solutions of ΔN-VAT, the enzyme itself is degraded in an ATP- and proteasome-dependent manner (*Figure 1a–c*). The 20S proteasome does not hydrolyze ATP during proteolysis (*Barthelme et al., 2014*) and therefore this observation demonstrates that ΔN-VAT complexes must unfold other copies of ΔN-VAT via a process that requires ATP hydrolysis prior to degradation by the proteasome. VAT's ability to unfold pyruvate kinase (*Figure 1b*) and other VAT particles is consistent with the finding that it can recognize substrates with an unstructured N or C terminus (*Gerega et al., 2005*) and does not require a specific marker or target sequence for initiating protein unfolding.

Cryo-EM of full-length VAT previously showed that the enzyme adopts a helical split-ring conformation in the presence of ADP (*Huang et al., 2016*). ΔN-VAT particles with bound ADP assemble into long fibrils (*Figure 1d*). Power spectra from these fibrils (e.g. *Figure 1d*, inset), as well as their real space appearance, suggest that they are composed of helical split-ring structures interacting with each other top-to-bottom in a poorly-ordered helical filament. The ~1/55 Å$^{-1}$ distance from the origin to the first layer line is consistent with the pitch of each NBD helix in the split-ring structure of VAT. These fibrils were not observed in the ADP-bound state of full-length VAT (*Huang et al.,*

**eLife digest** Proteins perform many important jobs inside cells. Each protein is made of a chain of building blocks called amino acids that fold together to form precise shapes. Old, damaged or poorly constructed proteins tend to be harmful to cells so it is important that they are recycled. Ring-shaped enzymes called AAA+ unfoldases help to recycle these proteins by unfolding their amino acid chains. However, since these enzymes only interact with their target proteins for a short time it has been difficult to find out what happens when the target protein binds to the enzyme.

AAA+ unfoldases use chemical energy provided by a molecule called ATP to drive protein unfolding. Ripstein et al. used a technique called electron cryomicroscopy to study a AAA+ unfoldase from a microbe called *Thermoplasma acidophilum*. The enzymes were supplied with a molecule called ATPγS, which is similar to ATP but releases its energy more slowly. Ripstein et al. used this slowed down reaction, as well as new computer algorithms to analyse the microscopy images, to examine the structure of the target bound to the enzyme.

The experiments show that proteins move through the centre of the enzyme's ring as they unfold. At any time the enzyme is attached to the amino acid chain of the target protein in several places, which allows it to pull the protein through the ring in a "hand-over-hand" motion. Energy from each ATP molecule drives the enzyme to release one contact with the amino acid chain and form a new contact slightly further along the chain.

It remains unclear how AAA+ unfoldases identify and bind to the proteins they unfold and how they work alongside other recycling proteins inside cells. AAA+ unfoldases have an essential role in the biology of cells, and contribute to cancer and several other human diseases. Therefore, understanding how these proteins work may aid the development of new drugs to treat these diseases.

*2016*), possibly because the NTDs block their formation due to steric hindrance. With the slowly-hydrolysable ATP-analogue ATPγS bound, full-length VAT exists in both the split-ring conformation and a six-fold symmetric stacked-ring conformation (*Huang et al., 2016*). Similarly, ΔN-VAT with bound ATPγS forms a mixture of fibrils (*Figure 1e*) as well as single particles, the majority of which appeared to be six-fold symmetric (*Figure 1e*, red circles). Single particle cryo-EM analysis was performed, excluding the fibrils by inspecting micrographs after automatic particle selection and manually deselecting particle images that were part of a filament. This process allowed calculation of a cryo-EM map of the six-fold symmetric stacked-ring conformation of VAT at 3.9 Å resolution. This substantial improvement over the 7.8 Å resolution obtained previously for the six-fold symmetric conformation of full length VAT (*Huang et al., 2016*) is likely due to a number of factors. First, removal of the NTDs in ΔN-VAT makes the complex more structurally homogeneous and allowed specimens to be prepared with thinner ice than could be used with full length VAT. Specimen preparation with gold grids likely reduced beam-induced movement and improved image quality. Finally, preparation of crowded specimens with ~700 particles per micrograph improved power spectra and allowed more accurate fitting of contrast transfer function parameters at high resolution. The map resolution enabled construction of an atomic model for the complex (*Figure 2*, *Figure 2—figure supplement 1*, *Supplementary file 1*). In the structure, a central pore runs along the six-fold symmetry axis of the complex through the center of the stacked NBD1 and NBD2 rings (*Figure 2a*, left, hexagon). While it is possible that removal of the NTDs affects the conformations of the NBDs, Fourier shell correlation (FSC) of the map with the earlier map of full-length VAT (*Huang et al., 2016*) with the NTDs excluded by masking reached 0.5 at 8 Å resolution, consistent with the 7.8 Å resolution of the full-length VAT map (*Figure 2—figure supplement 1e*). The stacked-ring structure of VAT resembles recent structures of the VAT homologue p97 in the fully ATP-loaded state (*Zhang et al., 2000*; *DeLaBarre and Brunger, 2003*; *Banerjee et al., 2016*), except for the orientations between the NBD1 and NBD2 domains in VAT protomers and the relative rotation of the NBD1 and NBD2 rings. All AAA+ unfoldases have conserved sequences that extend from the NBDs towards the center of the ring, forming 'pore loops' that are essential for unfolding substrate (*Gerega et al., 2005*) (*Figure 2—figure supplement 2*). In the six-fold symmetric VAT structure,

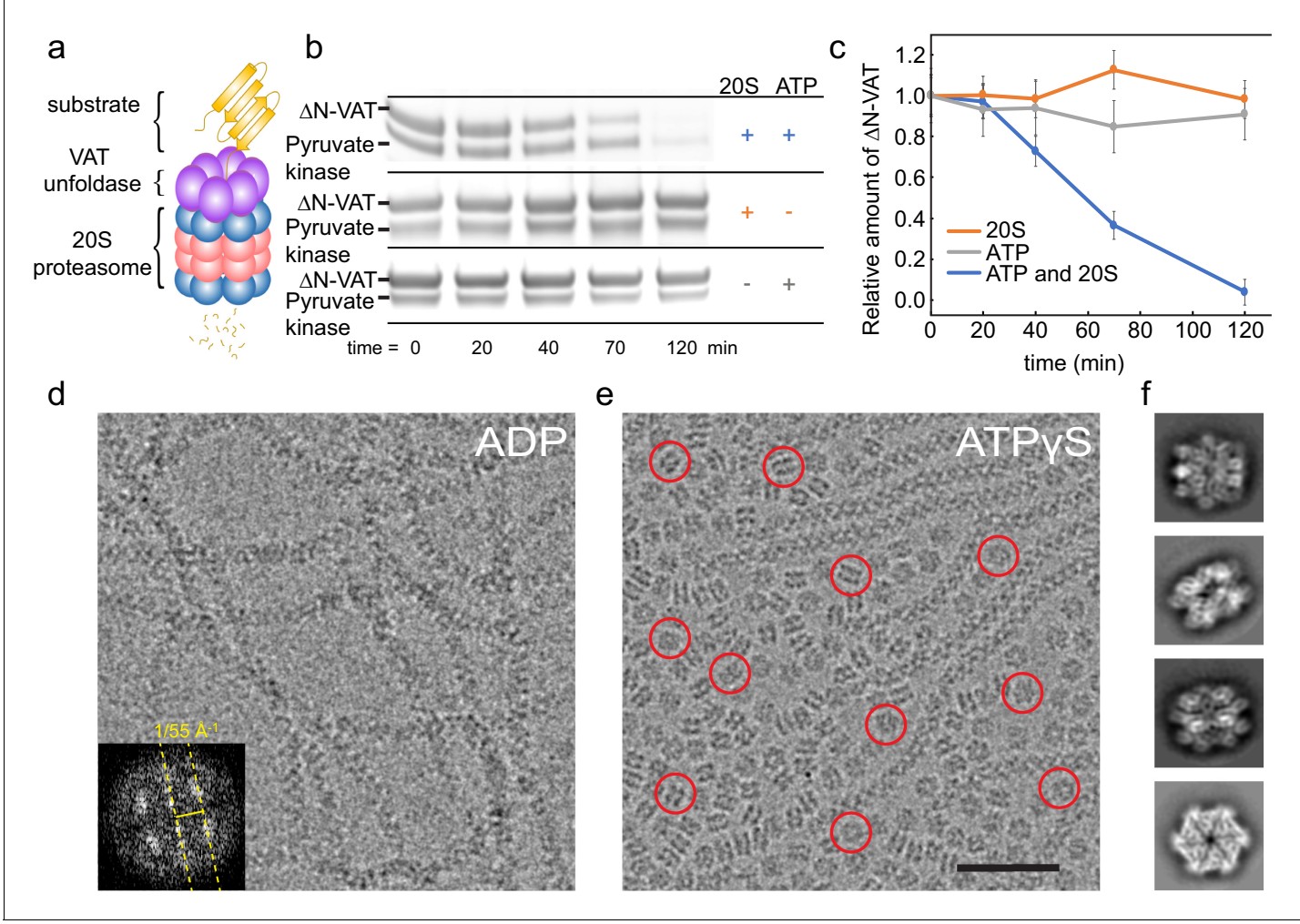

**Figure 1.** *Δ*N-VAT unfolds other copies of *Δ*N-VAT. (a) Cartoon showing VAT (purple ring) working in concert with the 20S proteasome (blue and pink rings) to degrade protein substrates (yellow). (b) An SDS-PAGE gel showing that both *Δ*N-VAT and pyruvate kinase (part of the ATP regenerating system) are degraded in an ATP and 20S proteasome dependent manner. (c) Quantification of the self-unfolding reaction. All measurements done in triplicate.(d) Representative section of a micrograph of *Δ*N-VAT in the presence of 5 mM ADP, corresponding to ~25% of the full field-of-view. Inset shows the power spectrum from a fibril, with a characteristic layer line at ~1/55 Å$^{-1}$. (e) Representative section of a micrograph of *Δ*N-VAT in the presence of 5 mM ATPγS. Example single particles are circled in red. Scale bar, 500 Å. (f) 2D class average images showing views of single particles of *Δ*N-VAT in the presence of ATPγS.

The following figure supplement is available for figure 1:

**Figure supplement 1.** Domain arrangement and oligomeric structure of VAT.

pore loop 1 of NBD1, containing conserved and essential tyrosine residues (Tyr264 and Tyr265) (*Gerega et al., 2005*), forms an opening that is ~28 Å across while pore loop 2 of NBD1 (residues 299 to 305) forms a slightly tighter aperture that is ~18 Å across (*Figure 2—figure supplement 2a*, left). Similar apertures are produced by the corresponding pore loop 1 of NBD2 (containing conserved residues Trp 541 and Val 542) and pore loop 2 of NBD2 (residues 575–579) (*Figure 2—figure supplement 2b*, left). An atomic model could be built for a significant fraction of the map (*Figure 2b*), including the linker region between NBD1 and NBD2 (*Figure 2c*, left), and both NBD1 and NBD2 have strong density for ATPγS in their nucleotide-binding pockets (*Figure 2c*, right) indicating that VAT adopts a rigid and symmetric conformation when fully loaded with nucleotide.

Unsupervised classification of ATPγS-loaded *Δ*N-VAT particle images using a new *ab initio* 3D classification algorithm (*Punjani et al., 2017*) allowed identification of a second conformation of the

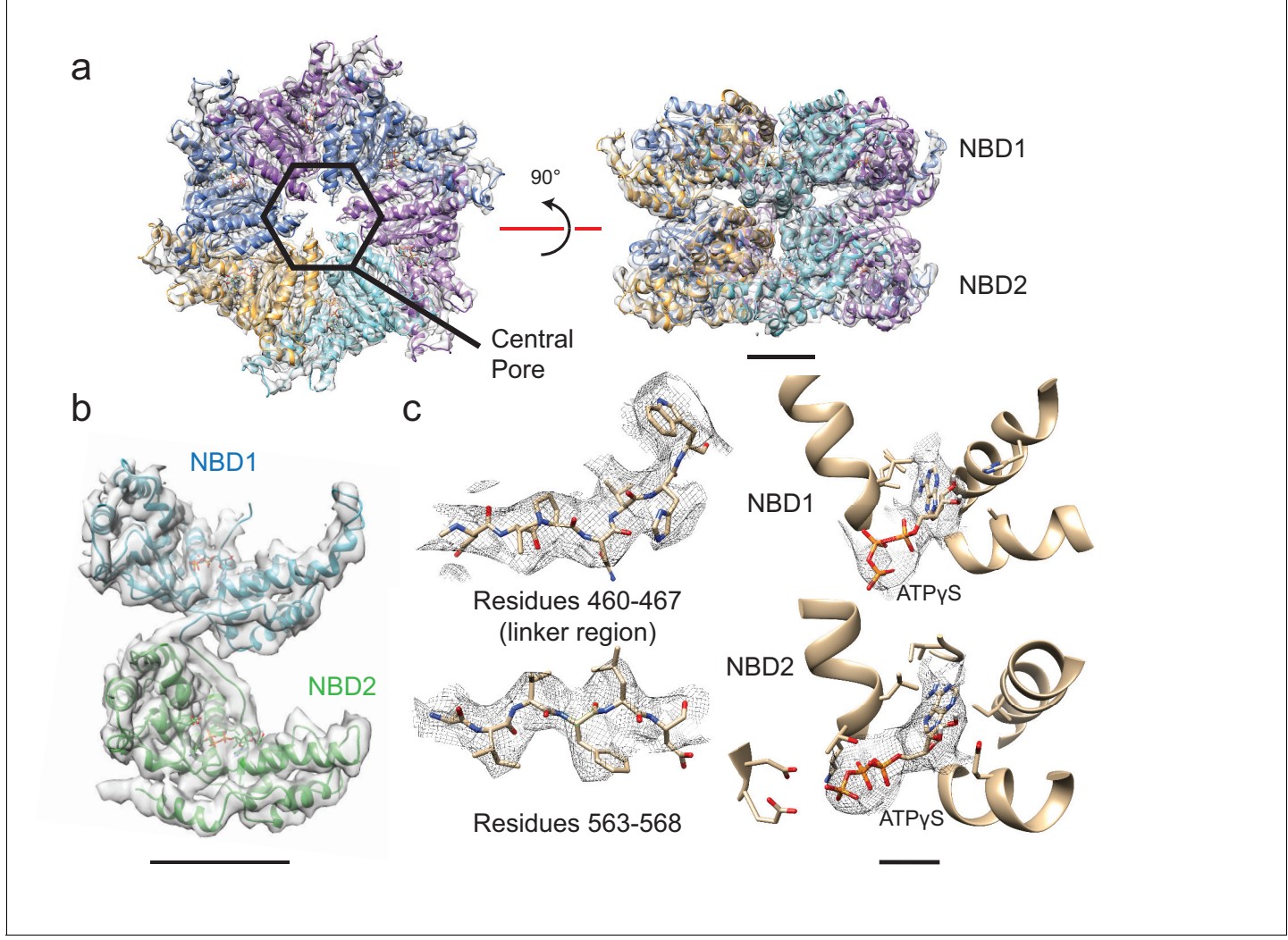

**Figure 2.** Structure of six-fold symmetric ΔN-VAT with bound ATPγS. (**a**) Density map and atomic model for ATPγS-bound ΔN-VAT at 3.9 Å resolution. A central pore runs through the middle of the complex. Scale bar, 25 Å. (**b**) Density map and model for a single protomer of VAT shows both AAA+ domains and the linker between them. Scale bar, 25 Å. (**c**) The cryo-EM density allowed identification of large amino acid side chains throughout ΔN-VAT, including in the linker region. Density corresponding to ATPγS is visible in all twelve catalytic sites. Scale bar, 5 Å.

The following figure supplements are available for figure 2:

**Figure supplement 1.** Cryo-EM map calculation and validation.

**Figure supplement 2.** Pore loops of VAT.

protein that was significantly different from known structures. The class corresponding to this conformation, containing ~15% of particle images, showed two ΔN-VAT complexes in close apposition to one another (*Figure 3*, *Figure 3—figure supplement 1*). Refinement of this structure to ~4.8 Å resolution revealed it to consist of a ΔN-VAT complex in the process of unfolding a neighboring complex as substrate (*Figure 3a*). Formation of this state may be dependent on the slow hydrolysis thought to occur with ATPγS. The map shows clear density for the substrate protein in an extended conformation as it runs almost directly through the central pore of the active ΔN-VAT (*Figure 3a and b*, red density). This substrate-bound conformation displayed a strong preference for particles to lie with their long-axes parallel to the plane of the specimen support, possibly because of the elongated shape of the complex (*Figure 2—figure supplement 1d*).

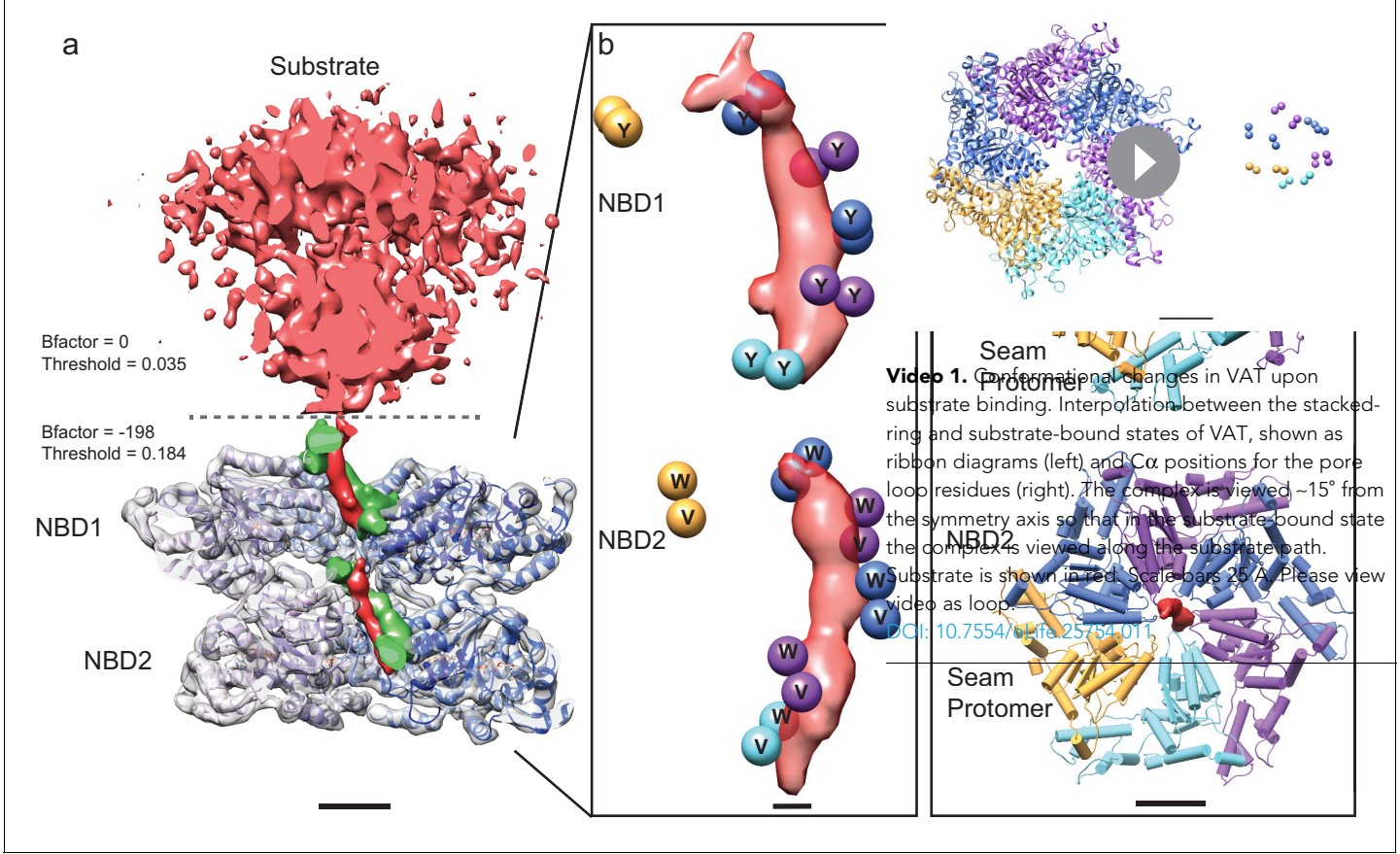

**Figure 3.** Substrate engagement by ΔN-VAT. (**a**) Cut-away side view of the density map and model for ΔN-VAT with substrate bound. The substrate complex is shown unsharpened in red above the dashed line. Density for the substrate-gripping loop residues is shown in green, with the enzyme-bound portion of the substrate shown as density (red) running through the center of the complex. Scale bar, 25 Å. (**b**) Positions of the Cα atoms of the pore loop residues (spheres) surrounding the experimental density for substrate (red). Five of the six protomers contact the substrate. Scale bar, 5 Å. (**c**) NBD1 and NBD2 rings with experimental density for substrate. Both NBD rings adopt similar conformations and a helical architecture, with five of the pore loops contacting the substrate. Scale bar, 25 Å.

The following figure supplements are available for figure 3:

**Figure supplement 1.** *Ab initio* classification to separate the six-fold symmetric and substrate-engaged states of VAT.

**Figure supplement 2.** Symmetry breaking and seam formation in the AAA+ rings of substrate-engaged VAT.

Density for the substrate above NBD1 of the active ΔN-VAT complex is weak and fragmented, and is most clearly visualized without sharpening of the map (***Figure 3a***, above dashed line) (***Zhao et al., 2015a***), suggesting that outside of the active enzyme the substrate has a flexible orientation relative to the pore of ΔN-VAT. The length of unfolded protein gripped by the pore of the active ΔN-VAT is ~80 Å, which corresponds to 12 to 14 extended amino acid residues. In both NBD1 and NBD2 five of the six VAT subunits form close contacts with the substrate through their pore loops (***Figure 3b and c***, and ***Figure 2—figure supplement 2a and b***, middle). The hydrophobic residues of these pore loops create a tight helical sheath around the substrate (***Figure 3a***, green). The top and bottom of the helix formed from the five subunits that grip the substrate are joined by a 'seam' subunit, which does not interact with the substrate and has protomer-protomer contacts that differ dramatically from the other subunits (***Figure 3c***, gold, ***Figure 2—figure supplement 2a and b***, middle). This helical arrangement of the pore loop residues maximizes the contact area between the unfolded substrate and the pore loops of the five subunits that form the helix (***Figure 3b***, ***Video 1***). Unfolded substrate that spans

the gap between the pore loops of NBD1 and NBD2 has a low density in the cryo-EM map, likely because of flexibility of the substrate in this region. This flexibility is likely due to a lack of tension on the substrate in the observed state. The transition from the stacked-ring to the substrate-engaged conformation of ΔN-VAT occurs with rearrangement of NBD1 relative to NBD2 in each protomer and a rotation of the NBD1 ring against the NBD2 ring (*Figure 3—figure supplement 2*, *Videos 1* and *2*). In the resulting asymmetric structure, the NDB1 and NBD2 of each protomer form similar interactions with their neighboring protomers. The two rings are also displaced horizontally relative to each other in the substrate-bound state and the central pore is tilted so that it passes between the rings at an angle of ~15° (*Figure 3—figure supplement 2c*). This rotation and displacement brings the NBD1 and NBD2 rings closer together than in the stacked-ring conformation, decreasing the distance between their pore loops.

In this study, only the newly identified conformation described above is found with substrate bound. Consequently, the six-fold symmetric stacked-ring state with ATPγS bound and the split-ring state with ADP bound (*Huang et al., 2016*) likely correspond to pre- and post-unfolding conformations of the enzyme. In the substrate-bound conformation, the seam protomer is positioned such that it does not interact with the substrate. Notably, recent structures of the 26S proteasome (*Chen et al., 2016*; *Wehmer et al., 2017*) also showed that either the Rpt5 or Rpt6 subunit in the hexameric Rpt ring of the 19S regulatory particle base is displaced from the pore, similar to the seam protomer in VAT (*Figure 3*). Furthermore, the translocation mechanisms of AAA+ DNA helicases such as E1, Rho, and MCM, also involve a helical gripping state in which only five of six protomers interact with a single DNA strand (*Lyubimov et al., 2011*; *Itsathitphaisarn et al., 2012*; *Enemark and Joshua-Tor, 2006*). The structure presented here suggests an analogous substrate translocation mechanism for VAT where the enzyme operates processively with a 'hand-over-hand' mechanism, shown schematically in *Figure 4a*, focusing on protomer 1 outlined in green. In the first step of this cycle (*Figure 4a*, top) protomer 1 is bound to unfolded substrate at the bottom of the helix of VAT protomers, protomer 2 in the seam position does not contact the substrate (*Figure 4a*, gold), and protomer 3 binds substrate at the top of the helix near the remaining folded protein. The nucleotide binding sites shared by the seam protomer and protomer 1 are empty (see below). As the seam protomer reengages with substrate at the top of the helix (step 2: *Figure 4a*, middle), protomer 1 disengages from the substrate and becomes the seam protomer, with protomer 6 now furthest along the unfolded substrate. The process repeats in a cyclical manner for additional protomers (e.g. step 3: *Figure 4a*, bottom, *Video 3*). The power stroke for substrate unfolding is provided by ATP hydrolysis, which occurs in sequence around the VAT ring, with ADP released by the protomer adjacent to the seam (counter clockwise from the seam when viewed from NBD1 toward NBD2, *Figure 4b*). Although the precise position where ATP hydrolysis and ADP release

occur in the cycle is not apparent from our data, it is clear that ATP must rebind to the seam protomer as it engages with substrate at the top of the protomer helix. As the cycle continues, each protomer pulls a single section of the substrate, corresponding to ~13 Å or approximately two extended residues, through the central pore until it becomes the lowest protomer in the helix and releases the substrate (*Video 3*). The cyclic nature of the ATP hydrolysis gives directionality to the substrate translocation, allowing unfolded protein to be passed to the proteasome for degradation.

The cyclic 'hand-over-hand' model requires that each protomer hydrolyze ATP in turn (*Figure 4b*). A single inactivated subunit would consequently have a severe effect on the rate of ATP hydrolysis, potentially stopping unfolding. To test the model, we measured the ATPase activity of VAT complexes that had been doped with different fractions (*r*) of catalytically-inactive

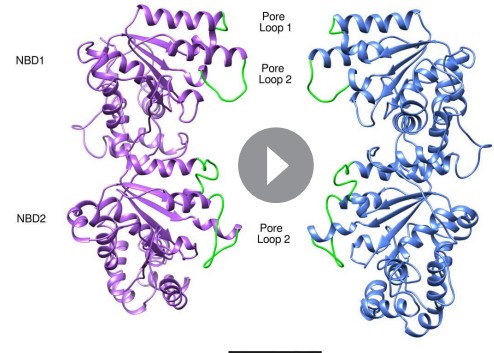

**Video 2.** Conformational changes in VAT protomers upon substrate binding. Interpolation between the stacked-ring and substrate-bound states of VAT, shown as a ribbon diagram for two VAT protomers. Substrate is shown in red. Pore loop residues are indicated in green. Scale bar, 25 Å. Please view video as loop.

protomers bearing mutations in the Walker B motifs of both NBD1 and NBD2 (E291Q, E568Q) (*Werbeck et al., 2008*) relative to wild type VAT ATPase activity (*A*). These experiments were carried out in the presence of GFP with an 11-residue ssRA tag as substrate (*Barthelme et al., 2014*). For multimeric enzymes where each subunit functions independently, ATPase activity is expected to decrease linearly as increasing amounts of catalytically dead mutant are incorporated randomly into the oligomers (*Figure 4c*, blue line). In contrast, the decrease in ATPase activity of the mixed complexes as a function of increasing concentration of catalytically dead mutant closely follows the $A = (1 - r)^6$ curve expected for the proposed hand-over-hand model (*Figure 4c*, gray line). The activity of VAT with increasing $r$ decreases faster than the $A = (1 - r)^6 + 5r(1 - r)^5$ curve (*Figure 4c*, dashed gray line) that is expected if ATP hydrolysis only stops when two of the six VAT protomers are inactivated. A similar curve is obtained when measuring the unfolding rate of the ssRA tagged GFP substrate rather than the ATPase rate (*Figure 4d*).

The slight elevation relative to the $A = (1 - r)^6$ profile in *Figure 4c and d* may reflect an 'escape' mechanism that allows stalled VAT complexes to disengage and then reengage with substrate, as seen with other molecular machines (*Werbeck et al., 2008*). In the absence of an escape mechanism ATP hydrolysis would proceed from one protomer to the next in a cyclical manner (*Figure 4b*) until a defective subunit is reached, preventing further hydrolysis and, thus, subsequent substrate unfolding. In contrast, if the machine is able to disengage at this point it would be possible for other functional subunits to hydrolyze ATP, although at reduced rates. The existence of an escape mechanism therefore has important implications for substrate unfolding, even in the context of wild type VAT, by allowing the release of stalled substrate. This release minimizes both the loss of active enzyme and increases catalytic efficiency. Our ATPase and unfolding assay data in *Figure 4c and d* cannot distinguish between a model where each protomer hydrolyzes ATP simultaneously or a sequential binding/hydrolysis mechanism as described in *Figure 4a and b*. However, previous ATPase measurements with VAT followed Michaelis-Menten kinetics (*Gerega et al., 2005*), with little cooperativity for ATP hydrolysis. This low cooperativity is inconsistent with simultaneous hydrolysis at all twelve nucleotide binding sites but is consistent with the proposed hand-over-hand mechanism, where only one NBD needs to bind and hydrolyze nucleotide at a time. The sequential model for ATP hydrolysis is also consistent with the asymmetry of the complex and nucleotide binding sites observed here, and the homology between VAT and other AAA+ enzymes known to hydrolyze ATP in a sequential fashion (*Enemark and Joshua-Tor, 2006*; *Kim et al., 2015*). ATP hydrolysis in the two NBDs has been shown to be coupled, with mutations in one domain also affecting ATP hydrolysis by the other (*Gerega et al., 2005*). The mechanism proposed here suggests that the two NBDs of each protomer move along the substrate in unison with each other (*Figure 4a*, *Video 3*). Both NBD1 and NBD2 of each protomer form similar interactions with their neighboring protomers. Preserving these similar interactions would require that they function in concert with each other. These conformations may be perturbed by the presence of the NTDs in full-length VAT. The conformations could also differ when the enzyme is in a physiological mixture of ATP, ADP, and inorganic phosphate, which could also affect the appearance of the bound substrate.

The similar conformations of both NDB rings suggests that the mechanism of substrate unfolding proposed here may well operate in single-ring AAA+ unfoldases as well, including $Rpt_{1-6}$ of the 19S proteasome, and the hexameric PAN and ClpX unfoldases. Thus, a picture emerges whereby AAA+ unfoldases adopt asymmetric substrate-bound conformations, creating narrow channels that sequester unfolded stretches of target proteins. Notably, even in molecular machines comprised of homo-oligomers, asymmetry can play an important functional role, in the case of VAT by priming the enzyme for processive translocation of its substrate.

## Data deposition

Electron microscopy density maps are deposited in the Electron Microscopy Databank under accession codes EMD-8658 and EMD-8659. Atomic models are deposited in the Protein Databank under accession codes 5VC7 and 5VCA.

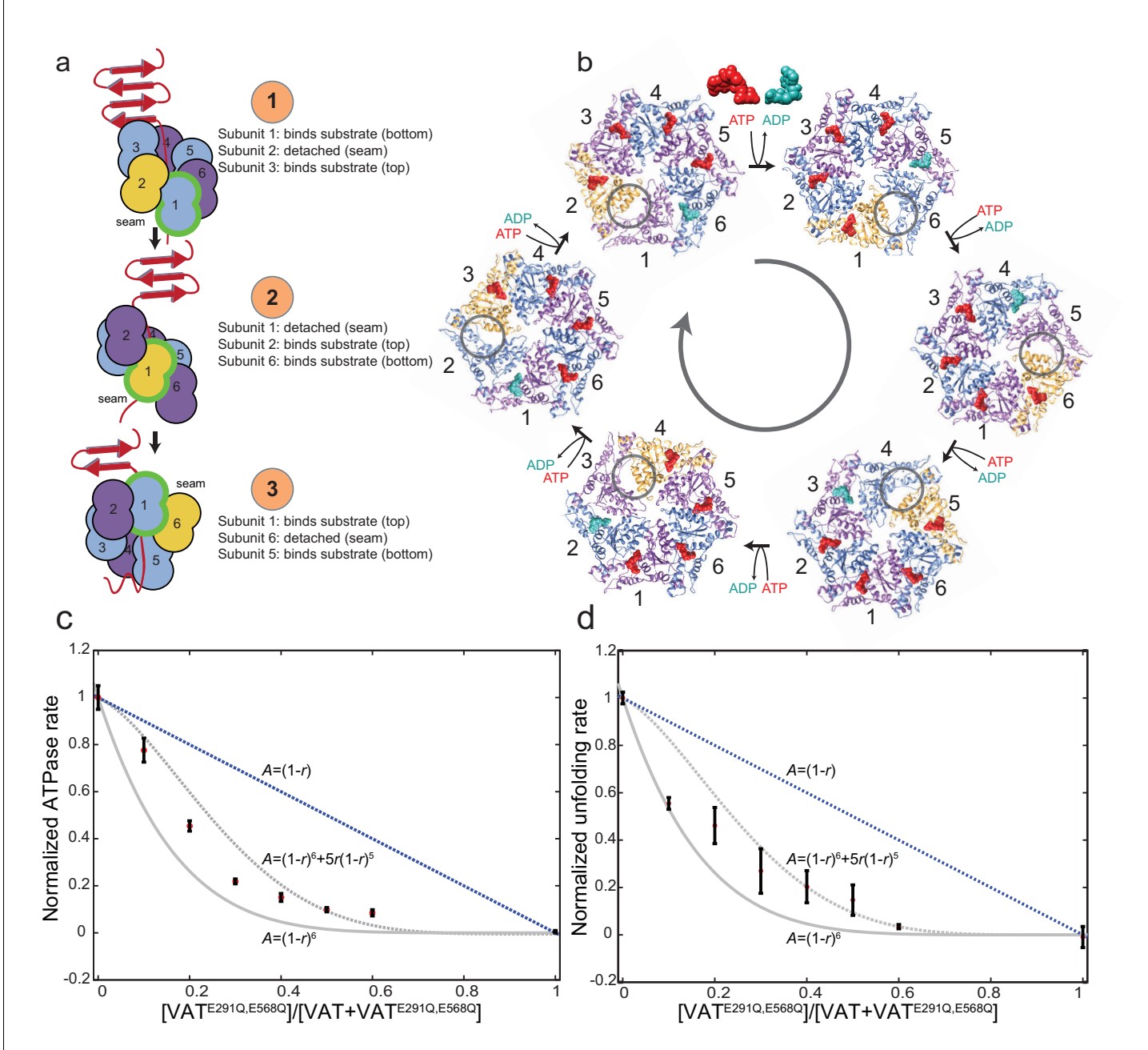

**Figure 4.** Model for substrate translocation by VAT. (**a**) The substrate-bound structure of ΔN-VAT suggests that the enzyme is processive, with a single subunit (green outline) that binds substrate at the lowest position (step 1) releasing as it becomes the seam subunit (yellow) (step 2) before re-engaging further along the substrate (step 3). Subunits are numbered 1–6. (**b**) The different conformations of VAT cycle around the hexamer, with ATP binding and ADP release with each transition. The seam subunit is indicated in yellow, with the adjacent subunits in the clockwise and counter-clockwise positions (viewed from NBD1 to NBD2) corresponding to protomers that are closest to and furthest from the folded substrate, respectively. (**c**) The ATPase activity $A$ of VAT with different ratios $r$ of catalytically inactivated VAT$^{E291Q,E568Q}$ mutant incorporated randomly into the enzyme hexamers. The activity decreases with increasing $r$ faster than the $A=(1-r)^6+5r(1-r)^5$ curve expected if two inactive mutants per complex were required to stop VAT activity, instead approximating the $A=(1-r)^6$ curve that is consistent with the proposed 'hand-over-hand' mechanism. Error bars, ±1 standard deviation. (**d**) The unfolding activity $A$ of VAT with GFPssra substrate, monitored by the decrease in fluorescence in the presence of ATP and a GroEL trap, with different ratios $r$ of catalytically inactivated VAT$^{E291Q,E568Q}$ mutant incorporated randomly into the enzyme hexamers. The activity decreases as in part c. Error bars, ±1 standard deviation.

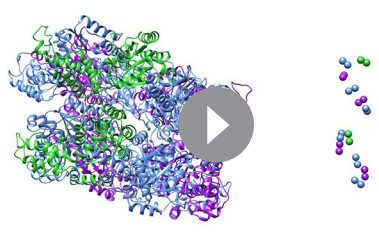

____

**Video 3.** The hand-over-hand model for substrate translocation. Interpolation between states where each subunit sequentially goes through the different possible positions in the helix. Each promoter's pore loop binds a section of substrate at the top of the helix and escorts that section toward the bottom of the image through five steps. On the sixth step the protomer releases the substrate and becomes the seam protomer, moving up and away from the central pore before reengaging at the top of the substrate. An arbitrary subunit is marked in green to help maintain orientation. The structures are shown as ribbon diagrams (left) and Cα positions for the pore loop residues (right). Scale bars, 25 Å. Please view video as loop.

## Materials and methods

### Protein expression and purification

An expression vector for the *T. acidophilum* VAT gene with the N-terminal 181 residues deleted (*ΔN-VAT*) (*Huang et al., 2016*) was prepared using a pProEx expression plasmid (Invitrogen, Carlsbad, CA). The construct included an N-terminal His$_6$-tag and a tobacco etch virus (TEV) cleavage site between the His$_6$-tag and the protein sequence. Expression and purification of *ΔN-VAT* were performed following the protocol described previously for full-length VAT (*Huang et al., 2016*). Walker B (catalytically inactivating) mutations (*Werbeck et al., 2008*) in VAT were made in a pET-28b vector that encoded codon-optimized VAT with an N-terminal His$_6$-tag and a tobacco etch virus (TEV) cleavage site between the His$_6$-tag and the *ΔN-VAT* sequence. The mutant protein was purified in the same way as wild type VAT and *ΔN-VAT* (*Huang et al., 2016*). To produce mixed complexes of wild type and catalytically inactivated VAT, different ratios of wild type and mutant protein were mixed and diluted to 15 µM (protomer) in unfolding buffer (50 mM HEPES pH 7.5, 200 mM NaCl, 6 M guanidine hydrochloride), and then refolded into hexamers by fast dilution at a ratio of 1:20 (v:v) into a refolding buffer (50 mM HEPES pH 7.5, 200 mM NaCl, 0.5 M arginine). After concentrating with a centrifuged concentrating device (Amicon, Millipore, Etobicoke, Canada ), refolded VAT mixtures were subject to size exclusion chromatography and the hexamer fractions were collected. Little or no aggregate or monomeric VAT was detected after refolding. The ADP- and ATPγS-bound forms of *ΔN-VAT* were prepared by incubation overnight at 25°C with apyrase (New England Biolabs, Whitby, Canada; 0.1 U/mg of *ΔN-VAT*). Apyrase was then removed by size exclusion chromatography and nucleotide was added to 5 mM.

### Electron cryomicroscopy

Purified *ΔN-VAT* at ~20 mg/mL in buffer (50 mM HEPES pH 7.5, 100 mM NaCl) was incubated with 5 mM ATPγS (90% pure, Sigma-Aldrich) for 5 min at room temperature before preparing cryo-EM grids. Immediately before grid freezing 0.05% (w/v) IGEPAL CA-630 (Sigma-Aldrich) was added to the protein solution to increase the number of particles adopting side views on the grid. Sample (2.5 µL) was applied to nanofabricated holey gold grids (*Marr et al., 2014*; *Meyerson et al., 2014*; *Russo and Passmore, 2014*), with a hole size of ~1 µm and blotted using a modified FEI Vitrobot for 5 s before plunge freezing in a liquid ethane/propane mixture (ratio of ~3:2) held at liquid nitrogen temperature (*Tivol et al., 2008*). Micrographs were acquired as movies with an FEI Tecnai F20 electron microscope operating at 200 kV and equipped with a Gatan K2 Summit direct detector device camera. Movies, consisting of 30 frames at two frames/s, were collected with defocuses ranging from 1.7 to 2.9 µm. Movies were obtained in counting mode with an exposure of 5 electrons/pixel/s, and a total exposure of 35 electrons/Å$^2$.

### Electron microscopy image processing

Whole frame alignment of movies was performed with *alignframes_lmbfgs* (*Rubinstein and Brubaker, 2015*) and the resulting averages of frames were used for contrast transfer function (CTF) determination with *CTFFIND4* (*Rohou and Grigorieff, 2015*) and automated particle image selection with *RELION* (*Scheres, 2015*). 851 particle images were selected manually and used to generate

two 2D class averages that served as templates for automated particle selection in *RELION*. Automated particle selection was followed by manual de-selection of non-protein features and protein from fibrils (*Figure 1d*). Particle coordinates were used to extract particle images in 256 × 256 pixel boxes from the unaligned movies, while performing individual particle movement correction and exposure weighting with *alignparts_lmbfgs* (*Rubinstein and Brubaker, 2015*). Magnification and CTF parameters were corrected for a previously measured 2% magnification anisotropy (*Zhao et al., 2015b*) resulting in a calibrated pixel size of 1.45 Å. A total of 171,381 candidate particle images were subjected to two rounds of 2D classification, ignoring the CTF until the first peak, in *RELION* (*Scheres et al., 2007*). Visual inspection and selection of 2D classes showing hexameric *Δ*N-VAT yielded 93,469 particle images, which were then transferred to the program *cryoSPARC* (*Punjani et al., 2017*) for *ab initio* 3D classification and refinement. Reference-free 3D classification runs using the stochastic gradient descent algorithm were performed multiple times with between two and six classes, before selecting two classes that yielded the highest quality maps. The best map for the stacked-ring conformation came from reference-free classification with two classes, while the best map for the substrate-engaged conformation came from reference-free classification with three classes. Only particle images that were confidently assigned to a class (*cryoSPARC* class probability ≥0.9) were used in refinement. With this threshold, gold-standard refinement of 75,205 particle images from the six-fold symmetric class, applying C6 symmetry, yielded a map at a resolution of 3.9 Å as measured by Fourier shell correlation (FSC). With the class probability threshold of 0.9, gold-standard refinement of 13,238 particle images from the substrate-engaged class with no symmetry applied yielded a map at a resolution of 4.8 Å by FSC. FSC curves were calculated with masks applied to remove solvent regions from the background of the maps. Masks are prepared automatically in *cryoSPARC* by binarizing the map, extending the edges of the resulting shape by a set distance, and tapering the mask to 0 over a second fixed distance with a cosine edge. Similar to *RELION* (*Scheres, 2012*), *cryoSPARC* measures and corrects for the effects of masking by randomization of high-resolution phases as described previously (*Chen et al., 2013*). The mask correction process introduces a dip in the corrected FSC at the resolution where randomization begins in the calculation. The final tight masking conditions were determined automatically by *cryoSPARC* by adjusting the extension and fading distances of the mask to optimize the reported resolution without introducing correlation from the mask.

## Map analysis and model building

Most of the density for loops, β-sheets and α-helices in the 3.9 Å resolution map were of sufficient quality to allow model building. A homology model of VAT that had been previously flexibly fit into a lower-resolution map (*Huang et al., 2016*) was rigidly fit into the map as individual domains (NBD1 and NBD2) with *UCSF Chimera* (*Goddard et al., 2007*). Final models were built with successive rounds of manual model building in *Coot* (*Emsley et al., 2010*) and real space refinement in *Phenix* (*Afonine et al., 2013*), and gave an *EMringer* score of 1.17, which is slightly better than the typical score of 1.0 for a map at this resolution. For the substrate-bound complex, the resolution was insufficient to determine side chain orientations, and a refined structure from the symmetric class was fit into the density by molecular dynamics flexible fitting (*MDFF* [*Trabuco et al., 2008*]). Density for each protomer was segmented with *UCSF Chimera* (*Goddard et al., 2007*; *Pintilie et al., 2010*). Only backbone atoms were subject to the fitting force during simulation, and optimization of geometry and rotamers was performed with *Phenix* (*Afonine et al., 2013*).

## Degradation assay

The auto-degradation of *Δ*N-VAT was measured following the protocol of Sauer and coworkers (*Barthelme et al., 2014*). Hexameric *Δ*N-VAT (0.3 μM) was mixed with an ATP-regenerating system (20 U·mL$^{-1}$ pyruvate kinase; 15 mM phosphoenolpyruvate) in the presence or absence of 10 mM ATP, with or without 0.9 μM *T. acidophilum* 20S proteasome. The reactions were carried out at 45°C with a total reaction volume of 50 μL (50 mM HEPES pH 7.5, 200 mM NaCl, 120 mM MgCl$_2$). Samples were withdrawn from the reaction at different times, quenched by heating in SDS-loading buffer, and subjected to SDS-PAGE. Intensities of the *Δ*N-VAT bands were analyzed with *Image Lab 4.0.1*.

## ATPase assays

Steady state ATP hydrolysis was measured with an enzyme-coupled assay described previously (*Nørby, 1988*). The reaction was carried out at a VAT concentration of 2.5 µM (protomer), and included an ATP regeneration system that contained 2.5 mM phosphoenolpyruvate, 0.2 mM NADH, 50 µg/mL pyruvate kinase, 50 µg/mL lactate dehydrogenase, 3 mM ATP, 200 mM NaCl, 120 mM MgCl$_2$, and 50 mM HEPES (pH 7.5). VAT complexes were prepared as mixtures of wild type and inactivated subunits (Walker B mutations) where Glu291 (NBD1) and Glu568 (NBD2) were substituted with Gln, followed by the unfolding/refolding protocol described above. Hydrolysis of ATP was monitored in a UV-visible light spectrometer at 340 nm, 25°C, in the presence or absence of 30 µM GFP substrate with an 11 residue ssRA tag (*Barthelme et al., 2014*). All measurements were done in triplicate. Similar results were obtained in the absence of substrate.

The expected activity of VAT complexes prepared by mixing different ratios of wild type and mutant protomers can be calculated as described previously (*Werbeck et al., 2008*). We define $r$ = [VAT$^{E291Q,E568Q}$]/[VAT]$_T$, which can be calculated from the concentration of mutant VAT protomers that are mixed with wild type, where VAT$^{E291Q,E568Q}$ is the VAT protomer bearing the E291Q/E568Q mutations and [VAT]$_T$ = [VAT$^{E291Q,E568Q}$]+[VAT$^{wild\ type}$]. We assume that the unfolding/refolding process leads to equal probabilities of insertion of either a mutant or wild type protomer into a VAT hexamer, so that $r$ is the probability of any protomer in the hexamer being a mutant. Under this assumption the fraction of VAT molecules with $k$ mutant protomers is given by

$$P(k) = \binom{6}{k} r^k (1-r)^{6-k} \tag{1}$$

where $\binom{n}{k} = \frac{n!}{k!(n-k)!}$, with the concentration of functional ATP hydrolysis sites in the protomer given by $2 \times (6-k)$ (two nucleotide sites per VAT). Assuming further that ATPase activity in the functional sites is not affected by the incorporation of inactive protomers until $m$ are added, in which case all hydrolysis stops, one can write the fractional activity of a solution of VAT complexes as a function of $r$, as

$$A(r) = \frac{1}{6} \sum_{j=0}^{m-1} (6-j) \binom{6}{j} r^j (1-r)^{6-j} \tag{2}$$

where $A(r)$ is activity normalized relative to the expected hydrolysis rate for fully wild type VAT hexamers at an identical concentration. Our structural model predicts that a single defective protomer is enough to eliminate ATPase activity in the hexamer. In this case $m = 1$ and $A(r) = (1-r)^6$. In contrast, in the case where there is no coupling between active and inactive protomers,

$$A(r) = \frac{1}{6} \sum_{j=0}^{6} (6-j) \binom{6}{j} r^j (1-r)^{6-j}$$
$$= \sum_{j=0}^{6} \binom{6}{j} r^j (1-r)^{6-j} - \frac{1}{6} \sum_{j=0}^{6} j \binom{6}{j} r^j (1-r)^{6-j}$$
$$= 1 - r \tag{3}$$

*Figure 4c* plots measured $A(r)$ versus $r$, as obtained experimentally, showing that ATP hydrolysis rates decrease rapidly with $r$. Although the dependence of $A$ on $r$ is slightly less steep than expected for a model where a single defective protomer completely eliminates ATP hydrolysis (see text) the experimental data points decrease more steeply than predicted for $m = 2$ (two defective protomers are required to eliminate hydrolysis).

## Unfolding assays

Unfolding activity was assayed by monitoring the loss of fluorescence of a substrate consisting of green fluorescent protein (GFP) with an 11-residue ssRA extension at its carboxy terminus (*Weber-Ban et al., 1999*). Briefly, VAT at a concentration of 5 µM of hexamer was mixed with 1 µM GFPssRA and 1.6 µM GroEL (D87K), which acts as a trap for unfolded GFPssRA, keeping it in an unfolded

state. Reactions were carried out in the presence of 100 mM ATP, 50 mM HEPES (pH 7.5), 200 mM NaCl, and 120 mM MgCl$_2$. Fluorescence of GFPssRA was monitored in a fluorescence plate reader (SpectraMax M5$^e$) with an excitation wavelength of 480 nm and emission wavelength of 520 nm. Initial slopes of the reaction were used to calculate the unfolding rate of GFPssRA. VAT complexes were prepared as mixtures of wild type and inactivated subunits (Walker B mutations) as described for the ATPase assay above.

## Acknowledgements

We thank Ali Punjani, Marcus Brubaker, and Suhail Dawood for advice and assistance with *cryo-SPARC*. ZAR was supported by a scholarship from the Natural Sciences and Engineering Research Council of Canada. LEK and JLR were supported by the Canada Research Chairs program. This research was funded by operating grant MOP-133408 from the Canadian Institutes of Health Research (LEK) and by discovery grant 401724-12 from the Natural Sciences and Engineering Research Council (JLR).

## Additional information

### Competing interests

LEK: Reviewing editor, *eLife*. The other authors declare that no competing interests exist.

### Funding

| Funder | Grant reference number | Author |
|---|---|---|
| Natural Sciences and Engineering Research Council of Canada | | Zev A Ripstein |
| Canadian Institutes of Health Research | MOP133408 | Lewis E Kay |
| Canada Research Chairs | | John L Rubinstein Lewis E Kay |
| Natural Sciences and Engineering Research Council of Canada | 401724-12 | John L Rubinstein |

The funders had no role in study design, data collection and interpretation, or the decision to submit the work for publication.

### Author contributions

ZAR, Performed protein purification, electron microscopy, image analysis, atomic model building, and enzyme assays, Wrote the manuscript with input from the other authors; RH, Created the ΔN-VAT construct , Performed protein degradation assays; RA, Created the E291Q, E568Q VAT mutant and assisted with the purification of this protein; LEK, Supervised research; JLR, Supervised research, Wrote the manuscript with input from the other authors

### Author ORCIDs

Zev A Ripstein, http://orcid.org/0000-0003-3601-0596
Lewis E Kay, http://orcid.org/0000-0002-4054-4083
John L Rubinstein, http://orcid.org/0000-0003-0566-2209

## Additional files

### Supplementary files

• Supplementary file 1. Cryo-EM data acquisition, processing, and atomic model statistics.

## Major datasets

The following datasets were generated:

| Author(s) | Year | Dataset title | Dataset URL | Database, license, and accessibility information |
|---|---|---|---|---|
| Ripstein ZA, Huang R, Augustyniak R, Kay LE, Rubinstein JL | 2017 | VCP like ATPase from T. acidophilum (VAT) - Conformation 1 | http://www.ebi.ac.uk/pdbe/entry/emdb/EMD-8658 | Publicly available at the EMBL-EMD Protein Data Bank in Europe (accession no. EMD-8658) |
| Ripstein ZA, Huang R, Augustyniak R, Kay LE, Rubinstein JL | 2017 | VCP like ATPase from T. acidophilum (VAT) - Substrate bound conformation | http://www.ebi.ac.uk/pdbe/entry/emdb/EMD-8659 | Publicly available at the EMBL-EMD Protein Data Bank in Europe (accession no. EMD-8659) |
| Ripstein ZA, Huang R, Augustyniak R, Kay LE, Rubinstein JL | 2017 | VCP like ATPase from T. acidophilum (VAT) - Conformation 1 | http://www.rcsb.org/pdb/explore/explore.do?structureId=5VC7 | Publicly available at the RCSB Protein Data Bank (accession no. 5VC7) |
| Ripstein ZA, Huang R, Augustyniak R, Kay LE, Rubinstein JL | 2017 | VCP like ATPase from T. acidophilum (VAT) - Substrate bound conformation | http://www.rcsb.org/pdb/explore/explore.do?structureId=5VCA | Publicly available at the RCSB Protein Data Bank (accession no. 5VCA) |

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
