## [Decision Letter]

Thank you for submitting your article "Structure of a AAA+ unfoldase in the process of unfolding substrate" for consideration by *eLife*. Your article has been reviewed by two peer reviewers, and the evaluation has been overseen by a Reviewing Editor and John Kuriyan as the Senior Editor. The following individual involved in review of your submission has agreed to reveal his identity: Alberto Bartesaghi (Reviewer #2).

The reviewers have discussed the reviews with one another and the Reviewing Editor has drafted this decision to help you prepare a revised submission.

Summary:

This manuscript presents the structure of two intermediate conformations adopted by the AAA+ unfoldase VAT protein hexamer with its N-terminal domain removed (DN-VAT) in the presence of ATPgS: the so-called "stacked-ring" C6-symmetric conformation at 3.9A resolution, and the "substrate-engaged" asymmetric conformation at 4.8A resolution. A lower resolution structure of the stacked-ring conformation for full-length VAT was presented before by the same authors. Here, the authors report the unexpected observation that they have captured a cannibalistic state with one ΔN-VAT caught as a substrate by another ΔN-VAT. This substrate-engaged conformation adopted by DN-VAT is reported here for the first time, and is used to suggest a "hand-over-hand" substrate unfolding mechanism through the central pore of an asymmetric DN-VAT hexamer, which resembles those of other AAA ATPases such as E1, Rho and MCM helicases. Overall, the work is well performed and the manuscript is clearly written, and should be of interest to a broad cross-section of the readership of *eLife*. Nevertheless, the editors and the reviewers have some significant concerns that need to be addressed before the manuscript can be considered further.

Major Concerns:

1) The assertion that DN-VAT in the presence of ADP assembles into long fibrils comprised of helical split-ring structures is inadequately supported. A single micrograph showing the fibrils (Figure 1) without additional supporting information seems insufficient to substantiate this claim. Why wasn't this phenomenon observed for full-length VAT in (Huang et al., 2016)? If the fibrils are indeed helical arrangements of DN-VAT split-ring hexamers, one could potentially get a structure out of them. At the very least, power spectra from the fibrils would show the periodicities present. Presenting some additional evidence and an extended discussion addressing these points will help substantiate this claim.

2) Regarding the structure determination of DN-VAT intermediates in the presence of ATPgS, the authors state that "the fibrils could be excluded easily when selecting single particle images for analysis" but provide no details on how this was done. Particle picking was presumably done in RELION and although there is no information on what templates were used, this procedure alone was probably not enough to discriminate single-particles from fibrils. Moreover, if Figure 1 shows a representative micrograph as stated in the legend, the authors should have ~700 particles per micrograph (171,381 particles / 246 micrographs) which seems like an excessively high number of particles given the size of the field of view of a single micrograph. This was followed by two rounds of 2D classification where presumably the fibril positions were eliminated but there are no details about what criteria or strategies were used for class selection to achieve this. During 3D classification: "Multiple 3D reference-free classifications were performed with between two and six classes using the stochastic gradient descent algorithm, before selecting two classes that yielded the highest quality maps.", were the two final classes obtained from different classification runs then? There are also inconsistencies between the total reported number of particles and fractions that went into each 3D class: 72,205 is 80% of 93,469 (not 85% as stated in Figure 3—figure supplement 1), and 13,238 is 14% of 93,469 (not 15% as stated in Figure 3—figure supplement 1) which will also leave ~5,000 particles unaccounted for. Regarding Figure 2—figure supplement 1, the authors should also comment on why the substrate-bound class is composed almost exclusively of side views (which is not the case for the stacked-ring class) and whether this bears any connection to the selection criteria used during 2D/3D classification and the potential inclusion of particles from the fibrils.

3) There is no comparison made between the new higher-resolution stacked-ring C6-symmetric structure and the author's own EMD-3435 published previously (Huang et al., 2016). Are these structures similar up to the lower common resolution of 7.8A? Did the absence of the N-terminal domain have any effect on the structures? Given that the imaging conditions used to obtain both structures were identical and the fact that the new structures were obtained from only half the number of micrographs (246 vs. 514), what were the factors that contributed to obtaining significantly better resolutions (~4 Å vs. ~7 Å) in the present report? It seems unlikely that the use of cryoSPARC alone would be responsible for such an improvement. Related to this, the 4.8 Å substrate-engaged intermediate shown in Figure 2—figure supplement 2 (middle column) and the 7 Å split-ring conformation (EMD-3436) (right column) show very similar structural features that don't seem to adequately reflect the reported large resolution difference. Was resolution assessed in the same way in both cases?

4) Quality of the reconstructions. Figure 2—figure – supplement 1 provides information on the quality of the reconstructions. Panels (A) and (B) show the FSC curves vs. resolution using various masks. However, there is no definition for loose, tight or corrected. Furthermore, the FSC curves for the "corrected" map contain sharp spikes. Can the authors elaborate on these points?

5) Physiological relevance of the structural models.

A) The structures here do not contain the N-terminal domains, which are substrate-interacting domains. Can the authors provide information on the substrate specificity and recognition for VAT and how the current structure reconciles with the substrate recognition requirements?

B) Furthermore, N-terminal domains have been shown to play important roles in regulating ATPase activity of NBD1 or NBD2 in other type II AAA proteins, the nucleotide bound states for NBD1 and NBD2 in the absence of N-domain might thus be different compared to full length proteins. Furthermore, whether N-domain has different effects on NBD1 and NBD2? The authors must discuss the potential effects of the N-domain deletion on the state and conformation of NBD1 and NBD2.

C) The lack of continuous density for the substrate in the central pore is surprising given the tight grips within the two separate hexameric rings (NBD1 and NBD2). The structure would thus suggest that the handing over of the substrate from NBD1 to NBD2 is somewhat un-coordinated. Can the authors comment on this? Could this be due to the presence of a single nucleotide (ATP-gS) instead of a likely mix of ATP, ADP and ADP+Pi under physiological conditions or could it be that the N-terminal domain plays a role in coordinating substrate recognition and translocation? The authors should comment on these points.

6) The relationship between NBD1 and NBD2. In other closely related AAA proteins such as p97 and NSF, NBD1 and NBD2 domains have been proposed to affect each other although the exact relationship is still under debate. Does the NBD1 affect NBD2 and vice versa in terms of nucleotide binding and hydrolysis? The structure of the substrate-engaged form reveals the relationship between the two NBDs within a single protomer. A detailed analysis of their relationship should be of great significance to the understanding of their mechanisms. Surprisingly, in the model presented in Figure 4, the authors only considered a single NBD domain.

7) What evidence is there that ATP-γ-S is not being slowly hydrolyzed in the cryo-EM structures?

---

## [Author Response]

*Major Concerns:*

*1) The assertion that DN-VAT in the presence of ADP assembles into long fibrils comprised of helical split-ring structures is inadequately supported. A single micrograph showing the fibrils (Figure 1) without additional supporting information seems insufficient to substantiate this claim. Why wasn't this phenomenon observed for full-length VAT in (Huang et al., 2016)?*

It is indeed interesting that fibrils were seen with ∆N-VAT with ADP but not full length VAT with ADP. We believe that steric hindrance from the NTDs prevents the fibrils from forming with full length VAT. We have added the following clarification in the main text:

“These fibrils were not observed in the ADP-bound state of full-length VAT (Huang et al., 2016), possibly because the NTDs block their formation due to steric hindrance.”

*If the fibrils are indeed helical arrangements of DN-VAT split-ring hexamers, one could potentially get a structure out of them. At the very least, power spectra from the fibrils would show the periodicities present. Presenting some additional evidence and an extended discussion addressing these points will help substantiate this claim.*

We thank the reviewer for this suggestion. We did indeed consider helical analysis when we first observed these fibrils. However, power spectra from the fibrils showed that they were ordered only to low resolution. We have added a power spectrum to Figure 1 and text to explain:

“Power spectra from these fibrils (e.g. Figure 1, inset), as well as their real space appearance, suggest that they are composed of helical split-ring structures interacting with each other top-to-bottom in a poorly-ordered helical filament. The ~1/55 Å^-1^ distance from the origin to the first layer line is consistent with the pitch of each NBD helix in the split-ring structure of VAT.”

*2) Regarding the structure determination of DN-VAT intermediates in the presence of ATPgS, the authors state that "the fibrils could be excluded easily when selecting single particle images for analysis" but provide no details on how this was done. Particle picking was presumably done in RELION and although there is no information on what templates were used, this procedure alone was probably not enough to discriminate single-particles from fibrils.*

We apologize for this lack of detail. We add the following text to the Results section to describe more explicitly how fibers were excluded:

“Single particle cryo-EM analysis was performed, excluding the fibrils by inspecting micrographs after automatic particle selection and manually deselecting particle images that were part of a filament.”

As well as the following in the Methods section:

“851 particle images were selected manually and used to generate two 2D class averages that served as templates for automated particle selection in *RELION*. Automated picking was followed by manual de-selection of non-protein features and protein from fibrils (Figure 1).”

*Moreover, if Figure 1 shows a representative micrograph as stated in the legend, the authors should have ~700 particles per micrograph (171,381 particles / 246 micrographs) which seems like an excessively high number of particles given the size of the field of view of a single micrograph. This was followed by two rounds of 2D classification where presumably the fibril positions were eliminated but there are no details about what criteria or strategies were used for class selection to achieve this.*

We did indeed pick ~700 particles per micrograph. The example images shown in Figure 1 are not a full micrograph but ~1/4 fields-of-view. The image of the ATPγS sample contains the expected ~175 particles for ~700 particles per micrograph. Showing a full micrograph in the figure does not illustrate particles clearly because of the limited resolution of computer monitors and printers.

We now clarify that this figure shows only part of a micrograph by modifying the caption of Figure 1:

**“**Representative section of a micrograph of ΔN-VAT in the presence of 5 mM ADP, corresponding to ~25% of the full field-of-view. e,Representative section of a micrograph of ΔN-VAT in the presence of 5 mM ATPγS.”

The reviewers are correct that we performed visual inspection of 2D classes to exclude classes that did not correspond to the overall size/shape of hexameric VAT. We have now included the following clarification in the Methods section:

“Visual inspection and selection of 2D classes showing hexameric ΔN-VAT yielded 93,469 particle images, which were then transferred to the program *cryoSPARC* (9) for ab initio3D classification and refinement.”

*During 3D classification: "Multiple 3D reference-free classifications were performed with between two and six classes using the stochastic gradient descent algorithm, before selecting two classes that yielded the highest quality maps.", were the two final classes obtained from different classification runs then?*

The two classes were indeed selected from separate runs. We have added the following to the Materials and methods section to clarify this point:

“Reference-free 3D classification runs using the stochastic gradient descent algorithm were performed multiple times with between two and six classes, before selecting two classes that yielded the highest quality maps. The best map for the stacked-ring conformation came from reference-free classification with two classes, while the best map for the substrate-engaged conformation came from reference-free classification with three classes.”

*There are also inconsistencies between the total reported number of particles and fractions that went into each 3D class: 72,205 is 80% of 93,469 (not 85% as stated in Figure 3—figure supplement 1), and 13,238 is 14% of 93,469 (not 15% as stated in Figure 3—figure supplement 1) which will also leave ~5,000 particles unaccounted for.*

We apologize for this confusion. Due to use of a threshold that only includes particle images in refinement that were confidently assigned to a 3D class, the number of particle images used in refinement is not the same as the number of particle images in a selected 3D class. We have clarified this point in Materials and methods section:

“Only particle images that were confidently assigned to a class (cryoSPARC class probability ≥0.9) were used in refinement. […] With the class probability threshold of 0.9, gold-standard refinement of 13,238 particle images from the substrate-engaged class with no symmetry applied yielded a map at a resolution of 4.8 Å by FSC.”

*Regarding Figure 2—figure supplement 1, the authors should also comment on why the substrate-bound class is composed almost exclusively of side views (which is not the case for the stacked-ring class) and whether this bears any connection to the selection criteria used during 2D/3D classification and the potential inclusion of particles from the fibrils.*

We do not believe that the prevalence of side views in the substrate-bound class is due to the selection criteria from 2D classification as the substrate-bound class emerged during 3D classification from the same dataset that gave the stacked-ring conformation. Instead, we think it is more likely that the elongated shape of two ΔN-VAT complexes side-by-side in the unfoldase-substrate complex leads to a preferred side view orientation for these particles. We have added the following text to the manuscript to suggest this possibility:

“This substrate-bound conformation displayed a strong preference for particles to lie with their long-axes parallel to the plane of the specimen support, possibly because of the elongated shape of the complex (Figure 2—figure supplement 1).”

*3) There is no comparison made between the new higher-resolution stacked-ring C6-symmetric structure and the author's own EMD-3435 published previously (Huang et al., 2016). Are these structures similar up to the lower common resolution of 7.8A?*

These maps are indeed the same out to the resolution limit of the earlier map, other than the presence of the NTDs. We include a new FSC curve in Figure 2—figure supplement 1 and added the text:

“Fourier shell correlation (FSC) of the map with the earlier map of full-length VAT (5) with the NTDs excluded by masking reached FSC=0.5 at 8 Å resolution, consistent with the 7.8 Å resolution of the full-length VAT map (Figure 2—figure supplement 1).”

*Did the absence of the N-terminal domain have any effect on the structures? Given that the imaging conditions used to obtain both structures were identical and the fact that the new structures were obtained from only half the number of micrographs (246 vs. 514), what were the factors that contributed to obtaining significantly better resolutions (~4 Å vs. ~7 Å) in the present report? It seems unlikely that the use of cryoSPARC alone would be responsible for such an improvement.*

We do indeed believe that the absence of the NTDs was the key determinant in reaching higher-resolution than before with the C6 structure of VAT. We do not think that cryoSPARC was the main contributor to the improved resolution. We enumerate the factors that we believe led to improved resolution in the following added text:

“This substantial improvement over the 7.8 Å resolution obtained previously for the six-fold symmetric conformation of full length VAT (Huang et al., 2016) is likely due to a number of factors. […] Finally, preparation of crowded specimens with ~700 particles per micrograph improved power spectra and allowed more accurate fitting of contrast transfer function parameters at high resolution.”

*Related to this, the 4.8 Å substrate-engaged intermediate shown in Figure 2—figure supplement 2 (middle column) and the 7 Å split-ring conformation (EMD-3436) (right column) show very similar structural features that don't seem to adequately reflect the reported large resolution difference. Was resolution assessed in the same way in both cases?*

Resolution was indeed assessed in the same way for both maps. Initially, we were also struck by the similarity in appearance of the 4.8 and 7.0 Å resolution maps. We calculated the local resolution of the maps, thinking that the pore loops shown in Figure 2—figure supplement 2 were perhaps at similar resolutions. While we now include these local resolution maps in Figure 2—figure supplement 1 the calculation revealed that the pore loops were not particularly disordered in either map. Subsequently, we low-pass filtered the 3.9 Å resolution 6-fold symmetric △N-VAT map to 4.8 and 7.0 Å (see Figure 5). This exercise showed that while it is easy to tell the 3.9 Å resolution map apart due to obvious bulky side chains on α-helices, it is quite difficult to distinguish a 4.8 Å resolution map from a 7.0 Å resolution map. Consequently, we conclude that the reason why the split-ring and substrate-engaged states of VAT’s pore loops in Figure 2—figure supplement 2 look similar is that at 7.0 and 4.8 Å resolution there are few differences in structural features to distinguish them.

Author response image 1.Six-fold symmetric △N-VAT map filtered to 3.9, 4.8, and 7.0 Å resolution:**DOI:**
http://dx.doi.org/10.7554/eLife.25754.016

*4) Quality of the reconstructions. Figure 2—figure supplement 1 provides information on the quality of the reconstructions. Panels (A) and (B) show the FSC curves vs. resolution using various masks. However, there is no definition for loose, tight or corrected. Furthermore, the FSC curves for the "corrected" map contain sharp spikes. Can the authors elaborate on these points?*

On reflection, we have realized that it is not necessary to include all of the diagnostic FSC curves that are provided by cryoSPARC. Instead, we now include just the tight-masked and corrected FSC curves and explain these curves in the methods section of the manuscript. In manuscripts where Relion is used it is standard to provide the corrected FSC.

NB: the spike in the corrected FSC curve is a well-known phenomenon that can also be seen in the corrected FSC curves from Relion.

We have added the following explanatory text:

“FSC curves were calculated with masks applied to remove solvent regions from the background of the maps. […] The final tight masking conditions were determined automatically by cryoSPARC by adjusting the extension and fading distances of the mask to optimize the reported resolution without introducing correlation from the mask.”

5) Physiological relevance of the structural models.

*A) The structures here do not contain the N-terminal domains, which are substrate-interacting domains. Can the authors provide information on the substrate specificity and recognition for VAT and how the current structure reconciles with the substrate recognition requirements?*

From numerous unpublished studies in the Kay laboratory, we know that VAT is able to unfold any substrate protein with an accessible N or C terminus, without requiring the equivalent of an ssRA tag like ClpX. The data presented in Figure 1 demonstrates this point by showing VAT unfolding VAT, but also unfolding the enzymes of the ATP regenerating system (e.g. pyruvate kinase) used in the experiment. We have added the following text to explain this lack of specificity:

“VAT’s ability to unfold pyruvate kinase (Figure 1) and other VAT particles is consistent with the finding that it can recognize substrates with an unstructured N or C terminus (Gerega et al., 2005) and does not require a specific marker or target sequence for initiating protein unfolding.”

While there have been some suggestions that the NTDs of VAT do indeed recognize an unidentified protein sequence, these suggestions remain speculative and we have added:

“The physiological function of the VAT NTDs remains unclear.”

*B) Furthermore, N-terminal domains have been shown to play important roles in regulating ATPase activity of NBD1 or NBD2 in other type II AAA proteins, the nucleotide bound states for NBD1 and NBD2 in the absence of N-domain might thus be different compared to full length proteins. Furthermore, whether N-domain has different effects on NBD1 and NBD2? The authors must discuss the potential effects of the N-domain deletion on the state and conformation of NBD1 and NBD2.*

We agree that it is possible that removal of the NTDs could affect the conformation or state of the NBDs. However, we can see no evidence that this has occurred. Therefore, we acknowledge this possibility in the text:

“While it is possible that removal of the NTD affects the conformations of the NBDs, Fourier shell correlation (FSC) of the map with the earlier map of full-length VAT (Huang et al., 2016) with the NTDs removed by masking reached FSC=0.5 at 8 Å resolution, consistent with the 7.8 Å resolution of the full-length VAT map (Figure 2—figure supplement 1).”

In discussing the similarities in the NBD1 and NBD2 rings at the end of manuscript, we also add the caveat:

“These conformations may be perturbed by the presence of the NTDs in full-length VAT.”

*C) The lack of continuous density for the substrate in the central pore is surprising given the tight grips within the two separate hexameric rings (NBD1 and NBD2). The structure would thus suggest that the handing over of the substrate from NBD1 to NBD2 is somewhat un-coordinated. Can the authors comment on this? Could this be due to the presence of a single nucleotide (ATP-gS) instead of a likely mix of ATP, ADP and ADP+Pi under physiological conditions or could it be that the N-terminal domain plays a role in coordinating substrate recognition and translocation? The authors should comment on these points.*

We believe that the lack of continuous density for the substrate requires that there is a lack of ‘tension’ between the regions gripped by NBD1 and NBD2. We do not think it necessarily indicates that the passing of substrate from NDB1 to NBD2 is uncoordinated.

We add the following comment:

“This flexibility is likely due to a lack of tension on the substrate in the observed state.”

However, we acknowledge that the complex is formed in the presence of a single nucleotide, which may give it a different conformation than if it were formed with a physiological mixture of ATP, ADP, and Pi. We point out this caveat by adding:

“These conformations may be perturbed by the presence of the NTDs in full-length VAT. The conformations could also differ when the enzyme is in a physiological mixture of ATP, ADP, and inorganic phosphate, which could also affect the appearance of the bound substrate.”

*6) The relationship between NBD1 and NBD2. In other closely related AAA proteins such as p97 and NSF, NBD1 and NBD2 domains have been proposed to affect each other although the exact relationship is still under debate. Does the NBD1 affect NBD2 and vice versa in terms of nucleotide binding and hydrolysis? The structure of the substrate-engaged form reveals the relationship between the two NBDs within a single protomer. A detailed analysis of their relationship should be of great significance to the understanding of their mechanisms. Surprisingly, in the model presented in Figure 4, the authors only considered a single NBD domain.*

We now add the following text to discuss the relationship between NBD1 and NBD2 further in terms of nucleotide binding and hydrolysis:

“ATP hydrolysis in the two NBDs has been shown to be coupled, with mutations in one domain also affecting ATP hydrolysis by the other (Gerega et al., 2005). […] Preserving these similar interactions would require that they function in concert with each other.”

Do explain how NBD1 and NBD2 interact with each other, we have added:

“The transition from the stacked-ring to the substrate-engaged conformation of ΔN-VAT occurs with rearrangement of NBD1 relative to NBD2 in each protomer and a rotation of the NBD1 ring against the NBD2 ring (Figure 3—figure supplement 2, Video 1 and Video 2). […] This rotation and displacement brings the NBD1 and NBD2 rings closer together than in the stacked-ring conformation, decreasing the distance between their pore loops.”

We apologize for the confusion about what we intended to represent in Figure 4. The ovals in the figure were meant to represent both NBDs, not a single NBD. We have updated the cartoon, replacing the oval shape with an hourglass shape, to indicate that it represents both NBD1 and NBD2.

*7) What evidence is there that ATP-γ-S is not being slowly hydrolyzed in the cryo-EM structures?*

We believe the ATP γS is indeed slowly hydrolyzed, and we refer to the compound as ‘slowly hydrolysable’ in the manuscript. We have also added the following text to emphasize that the ATP γS is likely slowly hydrolyzed, and this hydrolysis may be required for formation of the substrate-bound state:

“Formation of this state may be dependent on the slow hydrolysis thought to occur with ATPγS.”